# CA-LoRA: Adapting Existing LoRA for Compressed LLMs to Enable Efficient Multi-Tasking on Personal Devices

**Weilin Zhao**[1*], **Yuxiang Huang**[1*], **Xu Han**[1†], **Zhiyuan Liu**[2,1†],
**Zhengyan Zhang**[1], **Kuai Li**[3], **Chen Chen**[3], **Tao Yang**[3], **Maosong Sun**[1]

[1]Department of Computer Science and Technology, Institute for Artificial Intelligence,
 Beijing National Research Center for Information Science and Technology,
 Tsinghua University, Beijing, China.
[2]Quan Cheng Laboratory, Shandong, China.
[3]Tencent Machine Learning Platform Department, Tencent Inc, Beijing, China.
{zwl23,huang-yx21}@mails.tsinghua.edu.cn,{hanxu2022,liuzy}@tsinghua.edu.cn

## Abstract

Recently, there has been a demand to deploy Large Language Models (LLMs) on personal devices such as laptops and smartphones. These LLMs have different model variants when handling different tasks. However, personal devices have limited resources and require reduced storage overhead. To address this, there are two key methods available: the first is model compression, which compresses LLMs into smaller sizes; the second is LoRA, which can transfer an LLM to other tasks with very few parameters, avoiding the storage of multiple model variants in multi-task scenarios by only preserving LoRAs. However, our experiments show that directly combining these two methods yields sub-optimal performance. Considering that the open-source community has already contributed many LoRAs to LLMs, we propose to adapt these existing LoRAs from the LLMs to their compressed version and introduce a Compression-Aware LoRA (CA-LoRA) framework. We incorporate knowledge inheritance and recovery strategies to recover the lost knowledge caused by model compression. Experiment results demonstrate that CA-LoRA outperforms the vanilla LoRA methods applied to a compressed LLM and achieves comparable performance to the non-compressed LLM with existing LoRA modules. The source code of CA-LoRA is available at https://github.com/thunlp/CA-LoRA.

## 1 Introduction

In recent years, large language models (LLMs) with billions of parameters (Brown et al., 2020; Black et al., 2022; Chowdhery et al., 2023) exhibit more powerful and comprehensive abilities, especially in terms of cognition and embodiment (Lewkowycz et al., 2022; Nakano et al., 2021; Driess et al., 2023). To extend the LLMs to multi-task scenarios (Zhou et al., 2022; Sheng et al., 2023), parameter-efficient fine-tuning (PEFT) (Houlsby et al., 2019; Hu et al., 2022) has been widely used, where an LLM is frozen as a backbone among different tasks and then tiny tunable PEFT modules are injected to learn task-specific knowledge. Compared with conventional full-parameter fine-tuning (FT), where a single LLM is tuned into multiple task-specific variants, PEFT tunes much fewer parameters and has lower memory overhead in multi-tasking while achieving comparable performance (Ding et al., 2023). Among various PEFT methods, LoRA (Hu et al., 2022) is one of the most popular.

Despite the success of LLMs, deploying LLMs in real-world scenarios is an important issue. Online LLM serving is well studied (OpenAI, 2022; Google, 2023), and S-LoRA (Sheng et al., 2023) have explored the idea of serving thousands of LoRAs for thousands of tasks around

---

 * indicates equal contribution.
 † indicates corresponding authors.

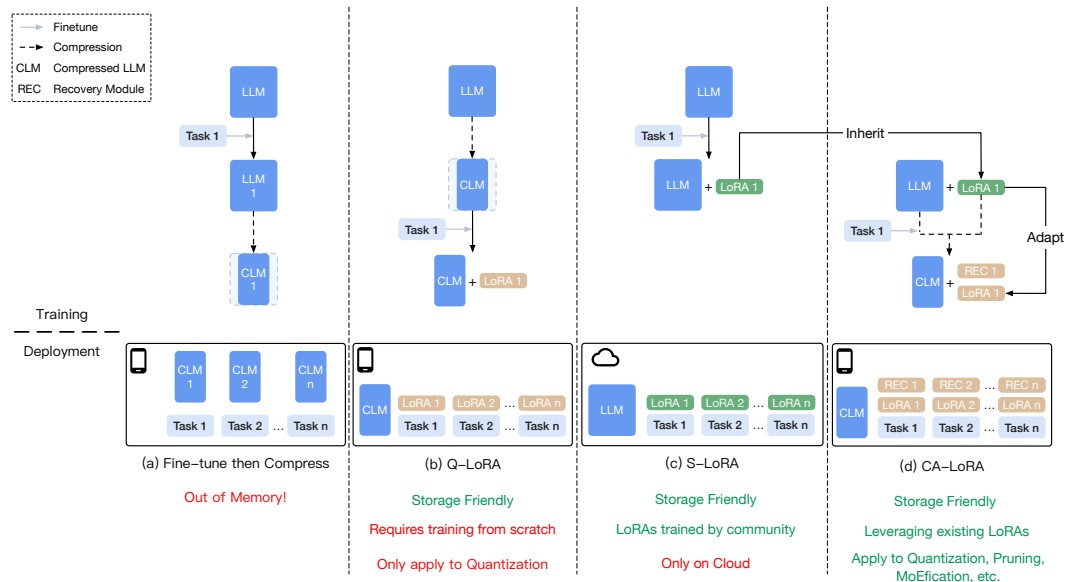

Figure 1: The training and deployment pattern of previous methods and our CA-LoRA.

a single LLM on a cloud server. However, how to deploy LLMs on personal devices, such as laptops and smartphones, is still in its early stages and faces many challenges (Ggerganov; team, 2023), as personal devices have severely limited storage resources compared with cloud servers.

To alleviate the storage constraint, model compression, such as quantization (Lin et al., 2024), pruning (Xia et al., 2024), and MoEfication (Zhang et al., 2022b), propose to compress an LLM into a smaller size, which we denote as the compressed LLM (CLM). Q-LoRA (Dettmers et al., 2023) makes a preliminary attempt in the scenario of multitasking on personal devices, which proposes to leverage the advantage of LoRA in the multi-task scenario by training LoRA on the compressed LLM. From our experiment, we found two shortages of this approach. Firstly, the LoRA training from scratch on the CLM is sub-optimal. Secondly, it only supports quantization, which has a limited model compression ratio and introduces storage overhead on personal devices.

Considering that the open-source community has already contributed many LoRAs to LLMs (HuggingFace), we propose to adapt those LoRAs from an LLM to its compressed version, we call it "Compression-Aware LoRA" (CA-LoRA). We introduce two mechanisms:

(1) **LoRA Knowledge Inheritance**. A stronger LLM can make learning LoRA modules easier. Meanwhile, the LoRA modules based on the stronger LLM can better grasp task-specific knowledge. Therefore, we propose to adopt the existing LoRA modules learned on the LLM as the initialization to learn the LoRA modules for its compressed version. In this way, the task knowledge of the LLM can be inherited to obtain more effective LoRA modules for the CLM, as shown in Figure 1.

(2) **Model Knowledge Recovery**. Whether LoRA modules trained on an LLM can work well with its compressed version is still an open problem, especially considering that model compression techniques may introduce knowledge loss and performance degradation. To recover the lost knowledge caused by model compression, we propose to inject a low-rank non-linear recovery module into the CLM to bridge the knowledge gap that arises from the compression process. We point out that compression techniques may weaken multiple capabilities of the LLM while only a small number of the capabilities it loses are required for a specific task. Therefore, the recovery module is also task-specific.

In experiments, CA-LoRA on CLM can recover the performance to the level of existing LoRA on LLM. These findings highlight the potential of CA-LoRA as an effective solution

for adapting existing multi-task ability from cloud servers to personal devices, thereby advancing the development of personal AI applications.

## 2 Related Work

This section mainly introduces PEFT and model compression. More details on LLMs can refer to surveys (Qiu et al., 2020; Han et al., 2021; Bommasani et al., 2021; Zhao et al., 2023).

### 2.1 Parameter-Efficient Fine-Tuning

Although LLMs can acquire rich knowledge from massive pre-training data to handle complex tasks in a zero-shot or few-shot manner (Brown et al., 2020; Black et al., 2022), there is still a need for adapting LLMs, to better stimulate the knowledge within LLMs to serve downstream tasks. For traditional PLMs, fine-tuning all the parameters of PLMs is the mainstream way to adapt them (Church et al., 2021), yet the parameter inefficiency makes this way costly to adapt LLMs (Ding et al., 2023).

To adapt LLMs to different tasks in a more efficient manner, various PEFT methods (Lester et al., 2021; Houlsby et al., 2019; Hu et al., 2022; Li & Liang, 2021; Ben Zaken et al., 2022) have been proposed. Specifically, LLMs are frozen and some model-independent tunable modules are injected into LLMs to help the adaptation process. PEFT modules are usually tiny so that PEFT methods can significantly reduce the cost of adapting LLMs. Among PEFT methods, LoRA is the most popular.

### 2.2 Model Compression

To improve the efficiency of LLM deployment, it is crucial to speed up the computation of LLMs. Model compression is a commonly used solution. Here we introduce task-agnostic model compression (Sanh et al., 2019) rather than task-specific compression (Sun et al., 2019), because task-specific compression is not memory-friendly in multi-task settings. Task-agnostic compression techniques include Quantization, Pruning, and MoEfication.

Quantization methods represent floating-point number LLM into fixed-point one, from 8-bit (Zafrir et al., 2019), 4-bit (Frantar et al., 2023; Lin et al., 2024) to 1-bit (Bai et al., 2021; Xu et al., 2024b). To avoid the performance degradation caused by quantization, quantization-aware training (QAT) (Stock et al., 2021) has been proposed to use a small amount of data to adjust the distribution of model parameters for quantization. Different from quantization methods that compress parameter representations, pruning methods directly discard some parameters. Commonly used pruning methods include structured pruning (Fan et al., 2020; Wang et al., 2020; Zhang et al., 2021; Xia et al., 2022; 2024) and unstructured pruning (Han et al., 2015; Chen et al., 2020; Xu et al., 2021). MoEfication (Zhang et al., 2022b), inspired by the mixture-of-experts (MoE) transformer (Lepikhin et al., 2021), aims to divide LLM parameters into multiple partitions, and each time only a few partitions are used to process input data. Typically, to make a compressed LLM behave the same as its original version, distillation objectives are often used to align the LLM and CLM, including aligning both output and intermediate states (Hinton et al., 2015; Sun et al., 2019; Jiao et al., 2020; Liu et al., 2022; Park et al., 2021). Zhang et al. (2022a) combines the above quantization, pruning, MoEfication and distillation into a mixed compression method, achieving a high compression ratio. More compression methods can refer to surveys (Liang et al., 2021; Xu & McAuley, 2022).

Recently, language models less than 3 billion parameters have been proposed (Bai et al., 2023; Javaheripi & Bubeck, 2023; Zhang et al., 2024; Hu et al., 2024; Liu et al., 2024b), while Xia et al. (2024); Zhang et al. (2022a) state that compressing LLM is better than small models.

Currently, combining LoRA with model compression is preliminary, and only some works attempt to combine LoRA with quantization (Dettmers et al., 2023; Liu et al., 2024a; Xu et al., 2024a). Recent work (Zhao et al., 2024) also attempts to conduct PEFT while compressing, but they are task-specific compression.

### 2.3 Multi-tasking

Maintaining multiple task-specific versions of an LLM in the storage is unacceptably resource-intensive (Zhou et al., 2022). As shown in Figure 1, $n$ full-model finetunes need to be done in vanilla fine-tuning for $n$ different tasks, posing significant challenges to the storage. In previous fine-tuning-then-compressing methods, this problem still exists.

Different LoRA modules can share a unified frozen LLM as their backbone, leading to lower computation and storage overhead in multi-task scenarios. PetS (Zhou et al., 2022) and S-LoRA (Sheng et al., 2023) use LoRA for multi-task serving on cloud servers. QLoRA (Dettmers et al., 2023) applies LoRAs on CLM, which is suitable for personal devices.

## 3 Methodology

### 3.1 Preliminary

For simplicity, we denote an LLM $\mathcal{M}$ as $\mathbf{Y} = f(\mathbf{X}; \boldsymbol{\theta}_{\mathcal{M}})$, where $f(\cdot)$ is the function of the whole transformer architecture, $\boldsymbol{\theta}_{\mathcal{M}}$ is the parameters of the LLM $\mathcal{M}$, $\mathbf{X}$ is the input, and $\mathbf{Y}$ is the output.

In the FT setting, all parameters of $\mathcal{M}$ (i.e., $\boldsymbol{\theta}_{\mathcal{M}}$) are tuned as follows

$$\boldsymbol{\theta}_{\mathcal{M}}^t = \arg\min_{\boldsymbol{\theta}_{\mathcal{M}}} \mathcal{L}(f(\mathbf{X}^t; \boldsymbol{\theta}_{\mathcal{M}}), \mathbf{Y}^t), \tag{1}$$

where $(\mathbf{X}^t, \mathbf{Y}^t)$ is the data of the downstream task $t$, $\mathcal{L}$ is the loss function of the task $t$. $\boldsymbol{\theta}_{\mathcal{M}}^t$ is the final task-specific parameters of the LLM $\mathcal{M}$.

In the LoRA setting, $\mathcal{M}$ is frozen, and additional LoRA modules $\mathcal{P}$ are tuned on task-specific data. We denote the parameters of the LoRA modules injected into $\mathcal{M}$ as $\boldsymbol{\theta}_{\mathcal{P}(\mathcal{M})}$. As shown in Figure 2, the computation of the transformer architecture is slightly changed due to the injected LoRA modules and becomes $\mathbf{Y} = f_{\text{LoRA}}(\mathbf{X}; \boldsymbol{\theta}_{\mathcal{M}}, \boldsymbol{\theta}_{\mathcal{P}(\mathcal{M})})$. The tuning process is formalized as

$$\boldsymbol{\theta}_{\mathcal{P}(\mathcal{M})}^t = \arg\min_{\boldsymbol{\theta}_{\mathcal{P}(\mathcal{M})}} \mathcal{L}(f_{\text{LoRA}}(\mathbf{X}^t; \boldsymbol{\theta}_{\mathcal{M}}, \boldsymbol{\theta}_{\mathcal{P}(\mathcal{M})}), \mathbf{Y}^t), \tag{2}$$

where $\boldsymbol{\theta}_{\mathcal{P}(\mathcal{M})}^t$ is the final task-specific LoRA modules collaborating with the LLM $\mathcal{M}$.

### 3.2 Framework

We propose a more efficient LoRA framework CA-LoRA, by adapting existing LoRA on an LLM into the compressed LLM (CLM).

As for the compression method, we choose task-agnostic model compression (Sanh et al., 2019) rather than task-specific compression (Sun et al., 2019) because task-specific compression is not memory-friendly in multi-task settings. After compressing the LLM $\mathcal{M}$, making $\mathcal{M}$ have fewer parameters or lower-bit representations, we denote the compressed LLM (CLM) and its parameters as $\mathcal{C}$ and $\boldsymbol{\theta}_{\mathcal{C}}$ respectively. Then the computation of the compressed model can be described as $\mathbf{Y} = f(\mathbf{X}; \boldsymbol{\theta}_{\mathcal{C}})$ without LoRA modules, and $\mathbf{Y} = f_{\text{LoRA}}(\mathbf{X}; \boldsymbol{\theta}_{\mathcal{C}}, \boldsymbol{\theta}_{\mathcal{P}(\mathcal{C})})$ with LoRA modules, where $\boldsymbol{\theta}_{\mathcal{P}(\mathcal{C})}$ indicates the parameters of the LoRA modules injected into the CLM $\mathcal{C}$. CA-LoRA can be formalized as

$$\boldsymbol{\theta}_{\mathcal{P}(\mathcal{C})}^t = \arg\min_{\boldsymbol{\theta}_{\mathcal{P}(\mathcal{C})}} \mathcal{L}(f_{\text{LoRA}}(\mathbf{X}^t; \boldsymbol{\theta}_{\mathcal{C}}, \boldsymbol{\theta}_{\mathcal{P}(\mathcal{C})}), \mathbf{Y}^t). \tag{3}$$

By combining model compression and LoRA, we can deploy a unified LLM to serve multiple tasks, while only maintaining tiny task-specific LoRA modules for each task. It is not difficult to imagine that adopting task-agnostic compression methods may weaken model performance, which will inevitably affect the search for the optimal parameters $\boldsymbol{\theta}_{\mathcal{P}(\mathcal{C})}^t$ and the effect of the final model $f_{\text{LoRA}}(\mathbf{X}; \boldsymbol{\theta}_{\mathcal{C}}, \boldsymbol{\theta}_{\mathcal{P}(\mathcal{C})}^t)$.

Inspired by the fact that model compression preserves those capabilities of LLMs that training a smaller model from scratch cannot master, we suppose that the LoRA modules

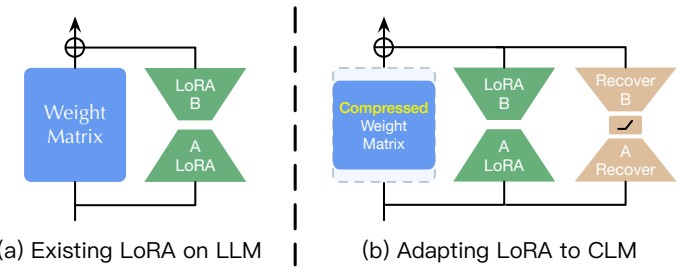

(a) Existing LoRA on LLM     (b) Adapting LoRA to CLM

Figure 2: The overall design of our CA-LoRA.

trained on the non-compressed LLM would contain certain task knowledge that the LoRA modules can hardly learn solely on the CLM. As shown in Figure 2, to better learn LoRA modules for the CLM, we adopt the method of inheriting the LoRA knowledge from those modules trained on the non-compressed LLM. To recover the lost knowledge caused by the compressing process, in addition to the LoRA modules $\mathcal{P}$, we add some knowledge recovery modules $\mathcal{R}$.

### 3.3 LoRA Knowledge Inheritance

Instead of training LoRA modules based on the CLM from scratch, we propose to use existing LoRA modules of the LLM as their initialization, then adapting them to the CLM. The adaption can lead to learning better LoRA modules on the CLM. Intuitively, it is more effective for a teacher to teach students the fundamentals of a discipline and then let students adapt their comprehension based on their circumstances rather than letting students learn from scratch. Formally, for task $t$, we first use existing LoRA module $\boldsymbol{\theta}^t_{\mathcal{P}(\mathcal{M})}$ trained on the non-compressed LLM $\mathcal{M}$ as the initialization of the LoRA module $\boldsymbol{\theta}_{\mathcal{P}(\mathcal{C})}$ of the CLM $\mathcal{C}$. After that, $\boldsymbol{\theta}_{\mathcal{P}(\mathcal{C})}$ is further tuned to $\boldsymbol{\theta}^t_{\mathcal{P}(\mathcal{C})}$ on the data of the task $t$ by using Eq. (3).

### 3.4 Model Knowledge Recovery

We propose to inject the knowledge recovery modules $\mathcal{R}$ into the CLM $\mathcal{C}$ to recover the lost of common knowledge during compression. As shown in Figure 2, we add a bypass next to each linear layer of the transformer architecture to add a small amount of change to the output states of those linear layers. To avoid introducing too many parameters, the recovery modules $\mathcal{R}$ adopt a low-rank structure like LoRA. Since LoRA is low rank while the compression of the model is often not of low rank, we have added additional non-linear activation in between to express some non-low-rank structures better. Formally, we denote the recovery module as

$$\mathcal{R}(\mathbf{X}) = \sigma(\mathbf{X}\mathbf{D})\mathbf{U}, \tag{4}$$

where $\mathbf{D}$ is the down projection matrix, $\sigma(\cdot)$ is the activation function, and $\mathbf{U}$ is the up projection matrix. $\mathbf{D}$ and $\mathbf{U}$ together form the parameter of the recovery module $\boldsymbol{\theta}_{\mathcal{R}}$.

To help obtain the optimal parameter $\boldsymbol{\theta}^t_{\mathcal{R}}$ of the recovery module for the task $t$, we design a distillation objective. Specifically, we select the existing LoRA modules trained with Eq. (2) and the LLM as the teacher. The whole distillation loss is given as

$$\mathcal{L}_{\text{DIST}}(\mathbf{X}^t; \boldsymbol{\theta}_{\mathcal{M}}, \boldsymbol{\theta}_{\mathcal{C}}, \boldsymbol{\theta}_{\mathcal{P}(\mathcal{C})}, \boldsymbol{\theta}_{\mathcal{R}}) =$$
$$\frac{1}{|\mathbf{X}^t|} \left\| f_{\text{LoRA}}(\mathbf{X}^t; \boldsymbol{\theta}_{\mathcal{M}}, \boldsymbol{\theta}^t_{\mathcal{P}(\mathcal{M})}) - f_{\text{LoRA}}(\mathbf{X}^t; \boldsymbol{\theta}_{\mathcal{C}}, \boldsymbol{\theta}_{\mathcal{P}(\mathcal{C})}, \boldsymbol{\theta}_{\mathcal{R}}) \right\|^2_2, \tag{5}$$

where $\mathbf{X}^t$ is the input data of the task $t$. Instead of first learning the recovery modules and then adding the inherited LoRA modules for further adaptation, we simultaneously conduct knowledge recovery and tune LoRA modules. The overall optimization problem

will become

$$
\begin{aligned}
\boldsymbol{\theta}^t_{\mathcal{P}(\mathcal{C})}, \boldsymbol{\theta}^t_{\mathcal{R}} =& \\
\arg \min_{\boldsymbol{\theta}_{\mathcal{P}(\mathcal{C})}, \boldsymbol{\theta}_{\mathcal{R}}} &\big[ \mathcal{L}(f_{\text{LoRA}}(\mathbf{X}^t; \boldsymbol{\theta}_{\mathcal{C}}, \boldsymbol{\theta}_{\mathcal{P}(\mathcal{C})}, \boldsymbol{\theta}_{\mathcal{R}}), \mathbf{Y}^t) \\
&+ \alpha \mathcal{L}_{\text{DIST}}(\mathbf{X}^t, \boldsymbol{\theta}_{\mathcal{M}}, \boldsymbol{\theta}_{\mathcal{C}}, \boldsymbol{\theta}_{\mathcal{P}(\mathcal{C})}, \boldsymbol{\theta}_{\mathcal{R}})\big].
\end{aligned}
\tag{6}
$$

Although CA-LoRA requires the non-compressed model to participate in the training process and results in extra training time, it is worth sacrificing little training cost for better task performance.

## 4 Experiments and Analyses

### 4.1 The Performance on Typical NLP Tasks

**Datasets** We evaluate CA-LoRA on 11 typical NLP datasets, more details are in Appendix A.

**Baselines and Implementation Details** We select T5-3b (Raffel et al., 2020) as the LLM and select its CLM from Zhang et al. (2022a), as shown in Table 1. Details about compressing T5-3b are in Appendix A. We compare the following 4 paradigms: (1) **T5-3b + LoRA**: LoRA modules are attached to the original T5-3b, and only the parameters of LoRA modules are tunable while the LLM is frozen. (2) **T5-base + LoRA**: Tunable LoRA modules are attached to the frozen T5-base model. (3) **CLM + LoRA**: LoRA modules are attached to the compressed versions of T5-3b, and then these CLMs are frozen and LoRA modules are tuned on task-specific data. (4) **CLM + CA-LoRA**: CA-LoRA is applied to the compressed versions of T5-3b, and only LoRA and recovery modules are tuned on task-specific data.

We set the bottleneck dimension of the LoRA modules and CA-LoRA recovery modules to 32 in all paradigms. The tunable parameters of LoRA and CA-LoRA modules are about 20M and 120M, respectively. More experimental settings are in our appendix.

Table 1: The models used in the experiments. The notation "T5-3b (X)" represents the T5-3b model with the setting "X". "M", "UP", "SP", and "Q" represent the model is compressed with MoEfication, unstructured pruning, structured pruning, and 8-bit quantization, respectively. "Q+UP+M" means "Q", "UP" and "M" are combined together to achieve higher compression ratios. "bf16" indicates the model is represented in bfloat16 floating-point rather than 32-bit floating-point format.

| Model | Model Size | Ideal Speedup |
|---|---|---|
| T5-3b (bf16) | 5.61 GB | 100% |
| T5-3b (M) | 3.74 GB | 150% |
| T5-3b (UP) | 2.81 GB | 200% |
| T5-3b (SP) | 2.81 GB | 200% |
| T5-3b (Q) | 2.81 GB | 200% |
| T5-3b (Q+UP+M) | 0.94 GB | 600% |
| T5-base (bf16) | 0.44 GB | 1400% |

**The Overall Performance**

Figure 3 shows that:

(1) The CLMs cannot perform as well as the original LLM, suggesting that task-agnostic compression leads to losing some task-specific knowledge. The performance of the compressed model may decrease due to the acceleration. If a mechanism makes up the performance gap without affecting the inference speed, applying compression will be more reasonable.

(2) CA-LoRA consistently outperforms vanilla LoRA methods. Such results indicate that task capabilities are effectively migrated through LoRA knowledge inheritance and model knowledge recovery mechanisms.

(3) Quantization and MoEfication have relatively little degradation on the model performance, and the degradation can be completely restored using CA-LoRA. However, the

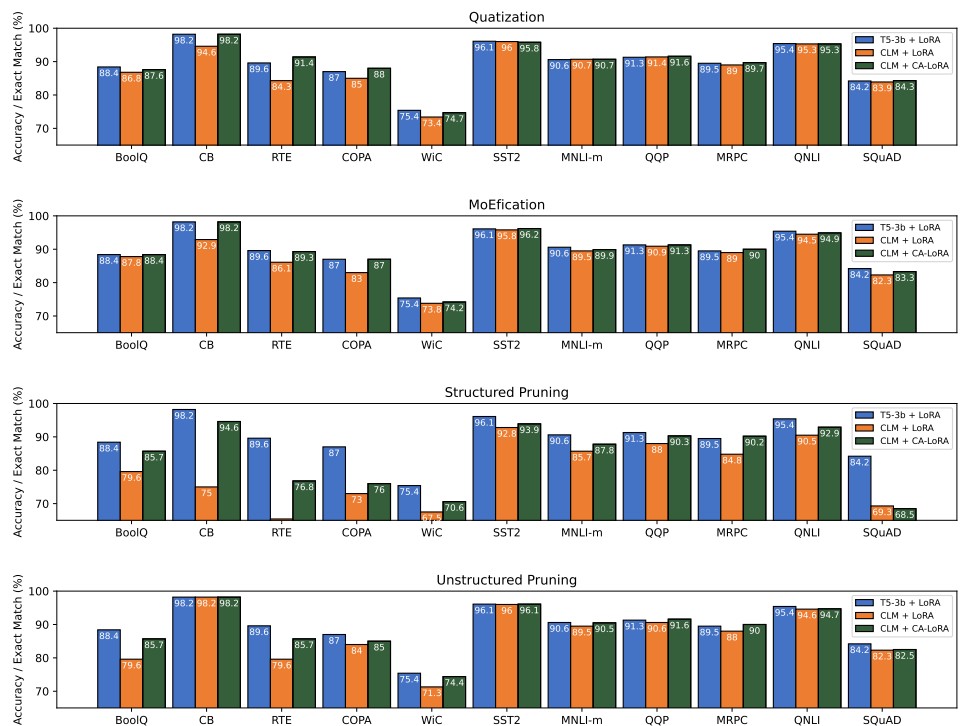

Figure 3: For typical NLP tasks, the results of LoRA and CA-LoRA on different CLMs (%).

Table 2: The results of applying CA-LoRA on the CLM T5-3b (Q+UP+M) that comes from the mixture of multiple compression methods (%).

| Method | Model Size(GB) | BoolQ Acc(%) | CB Acc(%) | RTE Acc(%) | COPA Acc(%) | WiC Acc(%) | SST2 Acc(%) |
|---|---|---|---|---|---|---|---|
| T5-3b + LoRA | 5.61 | 88.3 | 100.0 | 88.6 | 88.0 | 74.0 | 96.1 |
| T5-base + LoRA | 0.44 | 79.5 | 91.1 | 80.7 | 71.0 | 69.9 | 93.5 |
| CLM + LoRA | 0.94 | 85.2 | 89.3 | 82.9 | 80.0 | 70.5 | 94.8 |
| CLM + CA-LoRA | 0.94 | **87.1** | **100.0** | **84.3** | **86.0** | **72.0** | **96.2** |

| Method | MNLI-m Acc(%) | QQP Acc(%) | QQP F1(%) | MRPC Acc(%) | QNLI Acc(%) | SQuAD EM(%) | SQuAD F1(%) |
|---|---|---|---|---|---|---|---|
| T5-3b + LoRA | 90.6 | 91.3 | 90.7 | 89.5 | 95.4 | 84.2 | 92.5 |
| T5-base + LoRA | 84.8 | 90.6 | 89.9 | 86.5 | 93.1 | 79.0 | 87.8 |
| CLM + LoRA | 89.0 | 90.6 | 89.9 | **89.7** | 94.7 | 79.9 | **90.6** |
| CLM + CA-LoRA | **89.9** | **91.5** | **90.9** | 89.5 | **94.7** | **81.3** | 90.5 |

pruning methods cause more degradation, especially the structured pruning method. Even though, CA-LoRA recovers most of the performance loss caused by pruning methods.

To further evaluate CA-LoRA on a higher compression ratio, we use T5-3b (Q+UP+M) in Table 1, which is a CLM obtained from mixing quantization, MoEfication, and unstructured pruning, and has a close size to T5-base. Table 2 shows the experimental results. Results show that CA-LoRA is compatible and can be easily applied to a highly compressed model that uses multiple compression methods. Meanwhile, CA-LoRA demonstrates better performance on CLMs than training a small model with a close size from scratch.

### 4.2 The Performance on Instruction Tuning

The above experimental results have proven the strength of CA-LoRA for specific downstream tasks. In this section, we investigate the effectiveness of CA-LoRA when CA-LoRA is applied to a more general scenario, instruction tuning.

Table 3: The results of LoRA and CA-LoRA based on different compressed LLaMA (%). "$r_L$" and "$r_R$" respectively represent the intermediate rank of LoRA and Recovery modules. "†" represents that we couldn't reproduce QA-LoRA and the results are from the original paper.

| Method | $r_L$ | $r_R$ | MMLU STEM | Human | Social | Other | Mean | HumanEval | BBH | Drop | Overall Mean |
|---|---|---|---|---|---|---|---|---|---|---|---|
| LLaMa-13b | 0 | 0 | 36.4 | 44.9 | 54.1 | 53.1 | 47.0 | 17.7 | 36.6 | 35.0 | 34.1 |
| BMQuant (Zhang et al., 2022a) | 0 | 0 | 34.6 | 44.5 | 52.5 | 53.1 | 46.1 | 16.5 | 37.2 | 32.3 | 33.0 |
| NF4 (Dettmers et al., 2023) | 0 | 0 | 34.1 | 42.5 | 51.9 | 51.1 | 44.7 | 14.0 | 35.6 | 33.7 | 32.0 |
| QA-LoRA†(Xu et al., 2024a) | 0 | 64 | 38.3 | 48.4 | 54.9 | 55.2 | 49.2 | - | - | - | - |
| BMQuant+CA-LoRA | 16 | 16 | **37.6** | **45.1** | 54.3 | 54.1 | 47.6 | 17.7 | 35.8 | 33.3 | 33.6 |
| QLoRA (Dettmers et al., 2023) | 64 | 0 | 37.1 | 44.7 | **55.2** | 54.6 | **47.7** | 12.8 | **37.6** | 34.6 | 33.2 |
| QLoRA+CA-LoRA | 16 | 16 | **37.6** | 44.4 | 54.7 | **54.8** | 47.6 | **18.3** | 37.2 | 33.6 | **34.2** |

### 4.2.1 General Instruction Tuning

**Datasets** We select Alpaca (Taori et al., 2023) as the instruction tuning dataset and use 5-shot MMLU (Hendrycks et al., 2021) as the benchmark. We also use HumanEval (Chen et al., 2021), BBH (Suzgun et al., 2023), and Drop (Dua et al., 2019) for evaluation.

**Baselines and Implementation Details** Quantization is the core compression method for instruction tuning. Therefore, we compress LLaMA-13b (Touvron et al., 2023a) into an 8-bit quantization version for experiments, labeled "BMQuant", using the quantization method in Zhang et al. (2022a). We also add a 4-bit quantization NF4 baseline, which is the quantization method of QLoRA (Dettmers et al., 2023). Baseline methods are tested without instruction tuning on Alpaca. To evaluate CA-LoRA, we adopt 4 paradigms: QLoRA (Dettmers et al., 2023) and QA-LoRA (Xu et al., 2024a) for the previous compressing-then-tuning methods and BMQuant+CA-LoRA and QLoRA+CA-LoRA for our compression-aware tuning methods. More details of the LLaMA experiments can be found in Appendix B.

**The Overall Performance** Table 3 shows that quantization leads to overall performance degradation. BMQuant+CA-LoRA and QLoRA+CA-LoRA achieve better overall performance than other methods, indicating that the design of CA-LoRA is effective. QLoRA+CA-LoRA obtains better performance compared with QLoRA, showing the CA-LoRA can achieve **faster inference speed and better performance with fewer trainable parameters**.

### 4.2.2 Task-specific Instruction Tuning

**Datasets** We select CodeAlpaca-20k (Chaudhary, 2023) as the instruction tuning dataset and use 0-shot HumanEval (Chen et al., 2021) for evaluation.

**Baselines and Implementation Details** We compress LLaMa-2-13b (Touvron et al., 2023b) into an NF4 version, which is the quantization method of QLoRA (Dettmers et al., 2023). We then compare LoRA, QLoRA (Dettmers et al., 2023) and QLoRA+CA-LoRA. The intermediate rank of the LoRA and Recovery modules is set to 64. We use zero-shot evaluation of LLaMa-2-13b and its NF4 version as baselines.

**The Overall Performance** Figure 4 shows that QLoRA+CA-LoRA achieves better overall performance than baselines, indicating that the design of CA-LoRA is also effective in task-specific instruction tuning.

### 4.3 Ablation Studies

To more clearly demonstrate the inner mechanisms of CA-LoRA, we conduct ablation studies using the CLM T5-3b (Q+UP+M), as in Table 2, and adopt RTE for evaluation. For CA-LoRA, the intermediate ranks of LoRA and Recovery modules are 8. LoRA modules are injected into linear transformations of $\mathbf{W^Q}$ and $\mathbf{W^K}$ in attention layers. Recovery modules are injected into all linear transformations in all layers.

Table 4 shows how do different mechanisms in CA-LoRA help recover the knowledge loss caused by compression methods:

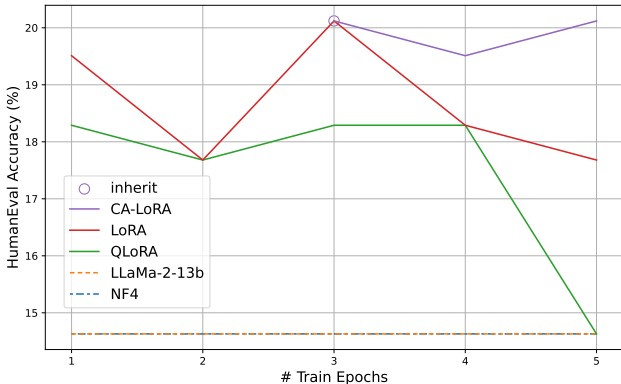

Figure 4: The performance of LoRA, QLoRA and CA-LoRA on HumanEval by instruction-tuning on CodeAlpaca-20k.

Table 4: The ablation studies of CA-LoRA on RTE (%). When eliminating inheritance, we keep the LoRA modules injected and train these modules from scratch. When eliminating recovery, we simply remove the recovery modules. When eliminating distillation, the training loss function only consists of the task loss.

| Inherit | Recover | Distill | RTE Acc |
|---------|---------|---------|---------|
|         |         |         | 83.6    |
| ✓       |         |         | 85.4    |
|         | ✓       |         | 82.5    |
|         |         | ✓       | 82.5    |
| ✓       | ✓       |         | 87.5    |
| ✓       |         | ✓       | 85.7    |
|         | ✓       | ✓       | 86.1    |
| ✓       | ✓       | ✓       | **88.6** |

(1) LoRA knowledge inheritance is effective. By initializing the tunable parameters with the existing LoRA modules trained on the non-compressed LLM, the performance of combining the final LoRA modules and the compressed LLM has been significantly improved. This indicates that in the optimization space of the compressed LLM, it is difficult to obtain the optimal task-specific parameters of LoRA modules based on random initialization, but more optimal LoRA modules can be achieved more easily using LoRA knowledge inheritance.

(2) Simply adding recovery modules brings a certain level of performance improvements in some circumstances. However, it can achieve further performance improvements by adopting our distillation strategy. This suggests combining recovery modules with the knowledge distillation strategy to enhance LoRA modules is necessary.

More ablations can be found in Appendix C.

### 4.4 The Convergence of CA-LoRA

Based on four CLMs of T5-3b (as in Figure 3), we test the convergence speed on BoolQ dataset. From Figure 5, we can find that due to our LoRA knowledge inheritance mechanism, CA-LoRA is superior to the vanilla LoRA methods in terms of convergence speed and final results. Adding the recovery module will not affect the convergence speed. Furthermore, when quantization, unstructured pruning, or MoEfication is used, the inheritance mechanism gives a better starting point for tuning LoRA modules. While in the case of structured pruning, even though the initial point of CA-LoRA does not work well on tasks, it is closer to the optimal point in the optimization space and converges faster.

Moreover, considering the existing LoRA modules based on the LLM may be trained by community users on their own private data and then uploaded to the Internet. When

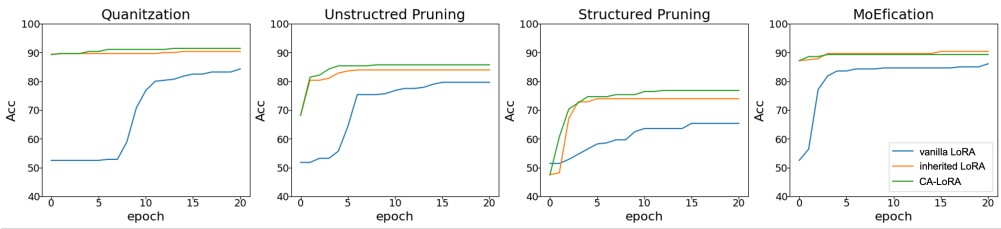

Figure 5: The convergence of vanilla LoRA, inherited LoRA (CA-LoRA without recovery modules), and CA-LoRA.

adapting these LoRA modules to the CLM, there may not be any task-specific data available for the adaptation process. In this case, we can directly use our LoRA inheritance mechanism without any fine-tuning while still achieving good performance on the CLM, when the compression method is quantization or MoEfication.

## 5    Conclusion

We propose an effective LoRA framework for adapting existing LoRA on LLM to the compressed LLM (named CA-LoRA) to deploy language models for multi-tasking on personal devices. Considering task-agnostic compression may cause losing task-specific knowledge, we introduce LoRA knowledge inheritance and model knowledge recovery to recover the lost knowledge. By inheriting the prior task knowledge of the LoRA modules learned on the non-compressed LLM, searching for the optimal LoRA modules for the compressed LLM becomes easier. Moreover, by introducing knowledge recovery modules to recover task-specific capabilities lost in the compression phase, collaborating LoRA modules with the compressed LLM can achieve comparable performance to those LoRA modules based on the non-compressed LLM.

### Limitations

Our work focuses on the efficiency and effectiveness, but it requires a small amount of extra training time. In this paper, we only choose LoRA as a representative of PEFT methods. In fact, our framework can be applied to any other PEFT methods. We focus on algorithmic support for running CA-LoRA on personal devices, but not the actual deployment implementation. The compression method adopted in this paper does not change the number of layers of the LLM. However, for compression methods that change the hidden dimensions of the model, how to transfer the knowledge of LoRA modules on the non-compressed LLM remains an open problem for our future work.

### Ethic Statement

This paper presents work whose goal is to advance the field of Machine Learning. There are many potential societal consequences of our work, none of which we feel must be specifically highlighted here.

### Acknowledgement

This work is supported by the National Key R&D Program of China (No.2022ZD0116312), National Natural Science Foundation of China (No. 62236004), and Institute Guo Qiang at Tsinghua University. Weilin Zhao is supported by Tsinghua University Initiative Scientific Research Program. Yuxiang Huang is supported by Tsinghua University Initiative Scientific Research Program (Student Academic Research Advancement Program).

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

## A   The Experimental Settings of Typical NLP Tasks

The typical NLP tasks to evaluate our methods including BoolQ (Clark et al., 2019), CB (De Marneffe et al., 2019), RTE (Bentivogli et al., 2009; Wang et al., 2019), COPA (Roemmele et al., 2011), WiC (Pilehvar & Camacho-Collados, 2018), SST-2 (Socher et al., 2013), MRPC (Dolan & Brockett, 2005), QQP (Wang et al., 2018), MNLI (Williams et al., 2017), QNLI (Rajpurkar et al., 2016), and SQuAD (Rajpurkar et al., 2016).

For compressed models, we use the compressed versions of T5-3b released by Zhang et al. (2022a), including 8-bit quantization, structured pruning, unstructured pruning, and MoEfication. Table 1 shows the models used in our experiments and their ideal inference speedup compared to T5-3b (Zhang et al., 2022a). Under this experimental setting, all paradigms are implemented with the open-source toolkit OpenDelta (Ding et al., 2023). We select the best learning rate among $\{1e-3, 5e-4, 1e-4, 1e-5\}$. The batch size is among $\{8, 16, 32, 64, 128, 256\}$. The weight decay is $1e-2$. The distillation coefficient in Eq. (6) is default to $\alpha = 0.05$.

## B   The Experimental Settings of Instruction Tuning

For Alpaca-LoRA and QLoRA, we test the released checkpoint. Alpaca-LoRA adds LoRA modules on every $\mathbf{Q}, \mathbf{K}, \mathbf{V}, \mathbf{O}$ projection and QLoRA adds LoRA modules to every linear transformation. In order to inherit from Alpaca-LoRA in QLoRA+CA-LoRA setting, the positions where LoRA modules are added are same as Alpaca-LoRA. For BMQuant+CA-LoRA  and QLoRA+CA-LoRA, we first apply the compression method to the backbone model, then we apply the LoRA modules and Recovery modules. We inherit from Alpaca-LoRA, then samely distill the modified model with Alpaca-LoRA. The positions of adding LoRA modules are the same as Alpaca-LoRA and Recovery modules are added to all linear transformations. The intermediate rank of additional modules are introduced in Table 3. All training settings follow QLoRA(Dettmers et al., 2023).

## C   More Ablation

**Is the improvement only caused by increased parameters?**

We carry out the study in Table 5. Each setting in this study maintains at least the same number of parameters as CA-LoRA. In CA-LoRA, the tunable parameters consist of LoRA modules and recovery modules. We introduce new settings of LoRA+LoRA and Large LoRA. More specifically, in LoRA+LoRA settings, we first inject LoRA modules to $\mathbf{W^Q}$ and $\mathbf{W^K}$ in attention layers, like conventional LoRA, and then inject extra LoRA modules to all linear transformations in both attention and feed-forward layers, replacing the recovery modules in CA-LoRA. Both LoRAs have an intermediate rank of 8. We then apply distillation to the modified model. In Large LoRA settings, we only inject LoRA modules with intermediate rank of 32 on $\mathbf{W^Q}$ and $\mathbf{W^K}$. From Table 5, we find that adding more parameters makes marginal improvements, showing that only adding more tunable parameters cannot bridge the knowledge gap caused by model compression.

Table 5: The studies controlling parameter quantity (%).

| Method | RTE Acc |
|---|---|
| LoRA | 83.6 |
| LoRA+LoRA | 85.4 |
| Large LoRA | 81.8 |
| CA-LoRA | **88.6** |

**Does adding more parameters slow down model inference?**

Since CA-LoRA introduces a little more parameters than conventional LoRA methods, it has a potential cause of slower inference. Considering the inference speed of compressed models is highly related to implementation, we evaluate CA-LoRA and conventional LoRA methods with the same backbone on the same platform to avoid uncertainties caused by backbone implementation. We report the average time and corresponding standard deviation of each model call. Table 6 and Table 7 show that the inference time of CA-LoRA and LoRA methods are almost the same, proving that inference bottleneck lies in the model itself, while the extra parameters added due to CA-LoRA do not slow down inference.

From the ablations above, we prove that each mechanism introduced in CA-LoRA is helpful to recover performance degradation of model compression without retarding model inference.

The structural design of CA-LoRA is reasonable, which is beneficial to knowledge inheritance and recovery.

Table 6: The average inference time on Alpaca dataset. The average time and corresponding standard deviations represent the time taken for each model call.

| Method | Avg. Time (ms) | Var. Time (ms) |
|---|---|---|
| QLoRA | 226.58 | 19.23 |
| QLoRA+CA-LoRA | 230.87 | 19.25 |

Table 7: The average inference time. "CLM" here represents the compressed T5-3b using mixture of compression methods. "# Param" represents the additional parameters compared with the backbone model. The average time and corresponding standard deviations represent the time taken for each model call.

| Method | # Param | Avg. Time (ms) |
|---|---|---|
| CLM+LoRA | 10M | (9761±17) |
| CLM+CA-LoRA | 60M | (9526±70) |

