# OpenReview forum: "CA-LoRA: Adapting Existing LoRA for Compressed LLMs to Enable Efficient Multi-Tasking on Personal Devices"
_colmweb.org/COLM/2024/Conference — COLM_

### Official Review · Reviewer_z39k · 2024-05-10

**Rating:** 7
**Confidence:** 2
**Ethics Flag:** 1

**Summary:**

This paper proposes a method to adapt a trained LoRA module for an LLM to create a LoRA module for it's compressed version by initializing the latter with the former before training, and learning an additional a task specific low rank recovery module to recover some of the knowledge lost from the base LM in the compression process. The technique is tested both for task specific adaptation and instruction tuning with T5-3B and adapters trained with the proposed method are shown to improve over standard LoRA for compressed versions of T5-3B obtained by quantization, MoEfication, structured pruning and unstructured pruning.

**Reasons To Accept:**

There is demand for efficient task specific models in many applications and this method shows promise as being an effective to perform task specific tuning of compressed LMs. The paper includes a result where on instruction tuning, compressed T5-3B + CA-LoRA outperforms T5-base + LoRA.

**Reasons To Reject:**

The experiments on this paper are restricted to only one model architecture - T5. This should be acknowledged as a limitation and potential direction for future work.
It is also unclear why the paper does not compare to QLoRA.

---

> ### Author Rebuttal · Authors · 2024-05-30
>
> Thanks for your valuable advice.
>
> ### W1: Restricted to T5
>
> We appreciate your suggestion to expand our experiments beyond the T5 model. While we do present results for Llama in Table 3, we acknowledge that our experiments currently have a narrow focus. In the revision, we will address this limitation and conduct further experiments on Llama to provide a more comprehensive analysis.
>
> ### W2: Does not compare to QLoRA
>
> We compare CA-LoRA and QLoRA in Table 3. The table shows that CA-LoRA can achieve comparable or even better performance than QLoRA. Owing to the addition of recovery modules, CA-LoRA can be combined with QLoRA and any other "compression + LoRA" works, showing greater compatibility and effectiveness. Furthermore, Table 6 shows that additional recovery modules do not reduce the inference speed, showing the efficiency of CA-LoRA.

---

### Official Review · Reviewer_rQV5 · 2024-05-13

**Rating:** 4
**Confidence:** 4
**Ethics Flag:** 1

**Summary:**

The main motivation is on device personalization where having the big LLM on in device memory is expensive. Hence the authors try to understand ways this requirement can be reduced. They propose using some existing compression techniques on the base LLM and then learning Lora modules on top of it. They propose two types of Lora modules one initialized from a task-specific Lora module and another that learns how to recover the information loss that happens due to compression. However, this part if very confusing to me.

**Questions To Authors:**

1. In CLM + LORA are the LORA modules initialized from the trained checkpoint as in CLM + CA-LORA.
2. The QA-Lora method is the strongest on MMLU and authors have omitted the results on it saying they took the results from the original paper. I feel like Table-3 should be completed with the results of QA-Lora on other benchmarks.
3. How are the recovery Lora modules initialized?
4. Can you provide training time comparisons CA-LORA with further training an inherited lora on top of a CLM? Because it seems like the cost overhead might be too high as distillation requires forward passes on two models.
5. Looking at Table 4, it seems like distillation is the least useful component, then it would be good to compare with inheret + recover at all the places in other tables? As it is not very clear how this combination performs at other places and if the additional training cost of distillation is worth it.
6. It does not conceptually make sense to me why do we need two lora modules in parallel to each other as the final output is just a linear combination of the two lora's anyways and both the lora's are trained on the same joint objective. Then why do we even need to recover lora. Can you just not have a single lora and use the objective in eqn-6 to obtain similar results? I feel like this would work equally well as the CA+LORA method.

**Reasons To Accept:**

Problem setting kind-off makes sense.

**Reasons To Reject:**

I have some issues with the experimental setting and the method that are listed below.

---

> ### Author Rebuttal · Authors · 2024-05-30
>
> Thanks for your valuable suggestion.
>
> ### Q6
>
> 1. A single LoRA trained with Eq.6 is evaluated in Tab.4 (Inherit + Distill), whose accuracy is 2.9 lower than our method. This means additional parameters (recovery module) for the recovery objective are necessary.
>
> 2. We further clarify that the recovery module with a non-linear activation function $\sigma$ is not a vanilla LoRA (Fig.2 and Eq.4), and it can represent a higher rank approximation. The compression loss does not have a low-rank pattern. For example, for unstructured pruning with 50\% sparsity, half of the parameters are eliminated, resulting in a large rank difference. To verify this, we also try "Inherited LoRA + Another LoRA + Distill" or "A single larger LoRA + Distill" in Appendix C (Table 5) and both of them have poor performance.
>
> 3. [1] found that multiple tiny LoRAs are more suitable for multi-task learning compared to a larger LoRA, with more stable training and better performance. Our two objectives can also be considered as a type of multi-task learning and require two separate  modules.
>
>
> ### Q1
> LoRA modules are randomly initialized for CLM + LoRA.
>
> ### Q2
>
> As in [2], developers fail to reproduce QA-LoRA, and QA-LoRA is limited to quantized LLMs, yet CA-LoRA can serve more compression methods like pruning or MoEfication. As how we build QLoRA + CA-LoRA (Table 3), adopting CA-LoRA to improve its orthogonal QA-LoRA is easy.
>
> ### Q3
>
> The recovery modules are randomly initialized.
>
> ### Q4
>
> We measure the training cost of CA-LoRA in Table 2 on RTE using $4\times$ A100. The results show that the training cost of CA-LoRA is about $1.6\times$ baseline, which is acceptable considering the performance benefit. Note that CLMs based on quantization and MoEfication are fast at decoding but slow at training.
>
> | Method | Time | GPU hours |
> |-|-|-|
> |CA-LoRA | 4197s | 4.66 |
> |CLM + LoRA| 2623s | 2.91 |
>
> ### Q5
>
> Thanks for pointing this out. Our main contribution is to inherit existing LoRA and bridge the CLM-LLM gap with an extra recovery module. Distillation is a commonly used method to match our purpose. We supplement some of the ablations based on Table 2 here for distillation, and we will include more analysis in our revision.
>
> | Compare to distill verion | BoolQ | CB | COPA | WiC |
> | - | - | - | - | - |
> | CLM+CA-LoRA w/o distill | -0.5 | -1.7 | -2.0 | -2.5 |
>
> [1] Wang et al. Multilora: Democratizing lora for better multi-task learning.
> [2] https://github.com/yuhuixu1993/qa-lora/issues/25

---

### Official Review · Reviewer_Zo74 · 2024-05-17

**Rating:** 7
**Confidence:** 4
**Ethics Flag:** 1

**Summary:**

The authors present CA-LoRA, a compression-aware parameter-efficient finetuning method based on LoRA. They train LoRA modules on an uncompressed LLM and then adapt them downstream to be used with a compressed version of the same LLM, with a relatively small number of additional parameters introduced in addition to the LoRA modules. They explore two model families -- T5 and Llama 2 -- across both SuperGLUE and instruction tuning tasks, and see noticeable improvements in nearly all scenarios they tested.

The experiments are reasonable, the idea is straightforward yet novel, and motivation is sound. The significance of this work lies in their stated motivation: enabling strong, static (or rarely updated) task-specific models to be deployed in resource-constrained environments, such as mobile devices. As far as I am aware, this is a novel method for adapting compressed language models to downstream tasks. They are also clear about limitations: this method requires extra compute to adapt the full sized model before being compressed. Whether this tradeoff is worth it depends on the specific training and deployment scenario.

The structure of the paper is good, but the clarity could be improved. For example, I am not sure why the extra parameters introduced are called “knowledge recovery modules” – based on my reading, they are essentially extra parameters (+ a nonlinear function) that allow the compressed model + LoRA weights extra freedom during finetuning, but they do not explicitly encode any knowledge from the original LLM. Additionally, I believe that many of the equations written are not vital to understanding the work, and replacing them with prose and/or diagrams would improve the overall readability of the paper.

**Questions To Authors:**

While the T5 experiments show value in CA-LoRA for task-specific finetuning, I believe the strength of the method lies in the instruction tuning experiments; I think they’re the most interesting and applicable to real world scenarios.

If compute allows, I have a few suggestions for extending the IFT experiments:
* Adding more evaluation domains, to cover other important properties of instruction-tuned LLMs:
  * Safety (XSTest, ToxiGen, etc)
  * Instruction-following ability (AlpacaEval 1 and/or 2, IFEval)
  * Math (GSM8K)
* If your compute budget allows, I would recommend adding another baseline:
  * Llama 2 13B finetuned (with LoRA, most likely), on Alpaca. This would be a fairer baseline vs your method than Llama 2 13B without instruction tuning.
* Try instruction tuning on a domain-specific IFT set, such as CodeAlpaca-20k, and then evaluating on domain-specific evaluations (e.g. for coding, HumanEval+ and MBPP+).

I don’t think any of these are strictly necessary, but I think they’d help strengthen your argument (especially the evaluations I mentioned in the first bullet point).

Additionally, if you have the space, I’d recommend moving Table 6 from the appendix to section 4.

Overall, I really like your work!

**Reasons To Accept:**

This paper demonstrates an interesting and useful strategy for building small but performant task-specific LLMs. They show applicability to real world use cases, and ground their work in an important motivation: deploying strong, task-specific models in resource constrained environments. Additionally, their method is theoretically straightforward, and adds little latency to inference. Overall, the method described seems like a very useful way to adapt compressed LLMs in certain scenarios.

**Reasons To Reject:**

The first experiments focus on older models and tasks, which are less relevant for real world scenarios as described in the motivation. SuperGLUE tasks are fairly outdated, and I would prefer to see the instruction tuning experiments expanded and highlighted, because those results are also promising.

The paper could also be written more clearly. The amount of equations is – in my opinion – unnecessary, and the paper would be more straightforward to read if some of these were changed to prose (and/or diagrams) instead.

---

> ### Author Rebuttal · Authors · 2024-05-30
>
> Thank you for your valuable suggestion.
>
> ### W1: IFT Experiments
>
> Conventionally, LoRA has been mainly used to build task-specific models. This has influenced our primary experiments, which are focused on tuning the T5 model for various specific tasks. Recently, with the advancements in the capabilities of LLMs, LoRA has also been employed in instruction tuning. We have recognized this emerging trend and conducted experiments using the Llama model and the instruction tuning dataset, Alpaca. Based on the results presented in Table 3, we believe that CA-LoRA can perform well on IFT, as evidenced by the existing experiments with Llama.
>
> We appreciate your insight into the limitations of our experiments and acknowledge the need to expand our instruction tuning experiments in the revision, particularly by incorporating more evaluation domains. Additionally, we will add more experimental results to make our existing conclusions more solid.
>
> ### Q1: Why do we name the recovery module “knowledge recovery module”
>
> Essentially, you are right that these additional parameters incorporating non-linear functions provide extra learning freedom during the tuning process. In CA-LoRA, LoRA can gain task-specific capabilities through inheritance, while the recovery module mitigates the loss of certain capabilities caused by model compression. Considering that our experiments have not conclusively proven model capabilities as equivalent to knowledge, we intend to refine our terminology to "recovery modules" in the revision for better clarity in our paper.
>
> ### About Writing
>
> Thank you for your valuable feedback on improving the writing and paper structure. We will take your suggestion into consideration and make the following changes in the next camera-ready version:
>
> 1. We will move Table 6 from the Appendix to Section 4 to show that CA-LoRA does not affect the inference speed.
>
> 2. We will remove unnecessary equations and add more figures to make our method clearer.

---

> > ### Comment · Reviewer_Zo74 · 2024-06-04
> >
> > Great, thank you for your detailed response! I believe the changes you've mentioned will make the paper stronger, and I'm very interested to see the additional evaluation results in the camera ready version.

---

### Decision · Program_Chairs · 2024-07-10

**Decision:**

Accept

**Comment:**

This paper combines two different efficiency methods, LORA and compression methods. The results are useful, but the experiments and domains are somewhat narrow.